# Do All Languages Cost the Same?
# Tokenization in the Era of Commercial Language Models

**Orevaoghene Ahia**◇ **Sachin Kumar**♠♡ **Hila Gonen**◇ **Jungo Kasai**◇
**David R. Mortensen**♠ **Noah A. Smith**◇♡ **Yulia Tsvetkov**◇

◇Paul G. Allen School of Computer Science & Engineering, University of Washington
♠Language Technologies Institute, Carnegie Mellon University
♡Allen Institute for Artificial Intelligence
{oahia,jkasai,nasmith,yuliats}@cs.washington.edu, sachink@allenai.org
hilagnn@gmail.com, dmortens@cs.cmu.edu

## Abstract

Language models have graduated from being research prototypes to commercialized products offered as web APIs, and recent works have highlighted the multilingual capabilities of these products. The API vendors charge their users based on usage, more specifically on the number of "tokens" processed or generated by the underlying language models. What constitutes a token, however, is training data and model dependent with a large variance in the number of tokens required to convey the same information in different languages. In this work, we analyze the effect of this non-uniformity on the fairness of an API's pricing policy across languages. We conduct a systematic analysis of the cost and utility of OpenAI's language model API on multilingual benchmarks in 22 typologically diverse languages. We show evidence that speakers of a large number of the supported languages are overcharged while obtaining poorer results. These speakers tend to also come from regions where the APIs are less affordable to begin with. Through these analyses, we aim to increase transparency around language model APIs' pricing policies and encourage the vendors to make them more equitable.

## 1 Introduction

Language models (LMs) have come to be known as general-purpose solutions capable of performing many tasks by following natural language instructions (Brown et al., 2020; Ouyang et al., 2022; Chung et al., 2022), and generalizing to new tasks at test time using a handful of demonstrations (Brown et al., 2020; Su et al., 2023). Motivated by their potential for commercial use, many industrial research institutions have moved away

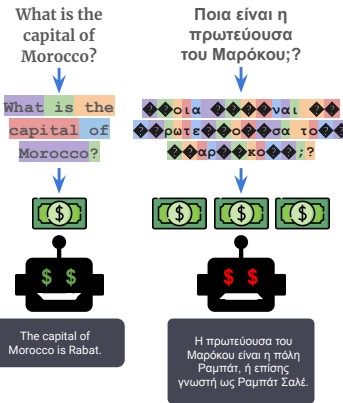

Figure 1: We investigate the effects of subword tokenization in LMs across languages with different writing systems. Our findings highlight disparities in the utility of LMs, as well as socio-economic disparities and increased costs in using commercial APIs for speakers of underrepresented languages.[1]

from openly releasing them (Abdalla et al., 2023). Instead, a new business model of LM as a Service (Sun et al., 2022) has emerged where LMs can be accessed for inference using (paid) web APIs. The majority of these models (Ouyang et al., 2022) offer multilingual capabilities, and the API providers charge the users proportionally to the number of tokens processed or generated.

In this work, we examine the fairness of this pricing model for different languages, based on how a "token" is defined in practice.[2] Most LMs rely on tokenizers that split text strings into chunks (subwords). Subword tokenizers (Sennrich et al., 2016; Kudo, 2018; Song et al., 2020) are typically data-driven and learn to split text based on frequency patterns of characters or bytes in some corpus. Prior work argued that, in multilingual settings, subword tokenizers lead to disproportion-

---

[1]OpenAI's tokenizer interface displays byte tokens absent from their vocabulary as "?".

[2]Code available at https://github.com/orevaahia/llm_tokenizer_cost

ate fragmentation rates for different languages and writing scripts (Zhang et al., 2022a; Rust et al., 2021; Muller et al., 2021). Many commercial LMs are multilingual, and text from languages that suffer from excessive fragmentation will be represented using more tokens. This directly increases cost of API usage for certain language speakers, even if they convey the same information as the others.

We highlight this unfairness through three stages of systematic analyses. First, we show evidence that tokenizers of popular LMs indeed over-fragment texts in certain language scripts and quantify the API cost disparity that this issue causes. We discover that the disparity is not caused just by data imbalance, but is rooted in the language properties or the ways they are represented in Unicode. Second, we show that languages with longer token lengths as a result of greater fragmentation derive less model utility with in-context learning (Brown et al., 2020). Finally, we find that languages that cost more and perform worse are often associated with populations of speakers for whom the APIs are less affordable on average, exacerbating the economic divide in the accessibility of NLP technology.

Through these analyses, we argue that commercial LM API vendors should revisit their processing and pricing strategies to be more equitable. In addition, we encourage the NLP community to pay better attention to tokenizers, an often neglected part of the LM pipeline.

## 2 Do All Languages Cost the Same?

### 2.1 Background

**Language Model APIs** Autoregressive LMs are trained to predict the next "token" given a previous context. Following the success of such models, many commercial LM web APIs have emerged and allow users to interface with the models using natural language instructions to perform various tasks with little to no exposure to the underlying workings of the models. The API providers often support dozens of languages and charge users[3] at a fixed rate based on the total number of input and generated tokens.[4] What constitutes a "token," however, is not a universally accepted definition but a design choice that the model developers make.

The total token count is also not immediately obvious to users except through a tokenizer interface[5] separate from the chat interface.

**Tokenization in LMs** Tokenization—segmenting text into atomic units—is an active research area. Proposed approaches range from defining tokens as whitespace-delimited words (for languages that use whitespace) which makes the vocabulary extremely large, to defining tokens as characters or bytes, making the tokenized sequences extremely long in terms of number of tokens; see Mielke et al. (2021) for a detailed survey. A commonly-used solution now is to tokenize text into *sub*word chunks. With Sennrich et al. (2016), one starts with a base vocabulary of only characters adding new vocabulary items by recursively merging existing ones based on their frequency statistics in the data. Other approaches judge subword candidates to be included in the vocabulary using an LM (Kudo, 2018; Song et al., 2021). For multilingual models containing data in a variety of scripts, even the base vocabulary of only characters (based on Unicode symbols) can be very large with over 130K types. Radford et al. (2019) instead proposed using a byte-level base vocabulary with only 256 tokens. Termed byte-level byte pair encoding (BBPE), this approach has become a de facto standard used in most modern language modeling efforts (Brown et al., 2020; Muennighoff et al., 2022; Scao et al., 2022; Black et al., 2022; Rae et al., 2022; Zhang et al., 2022b). In this work, we investigate the impact this tokenization strategy has on LM API cost disparity as well as downstream task performance (i.e., utility) across different languages.

### 2.2 Investigating the Impact of Byte-level Subword Segmentation

There are hundreds of distinct writing systems in the world (Hockett, 1997). BBPE, by design, makes vocabulary construction script-agnostic, allowing (in principle) new scripts to be supported later on without modifying the vocabulary. However, not only are different scripts encoded differently, their distribution in the training corpora varies widely. To investigate the effects of this variation, we propose the following research questions as the main focus of this work.

---

[3]While most services also have free tiers, they limit daily usage to a small number of tokens.

[4]E.g. see OpenAI models' cost: https://openai.com/pricing.

[5]https://platform.openai.com/tokenizer

**RQ1 (number of tokens): do all languages convey the same information with the same number of tokens?** We analyze the fragmentation of sequences in different languages with different tokenizers. We find that among the supported languages in popular LMs, there is a large variance in the average number of tokens required to convey the same information with some languages requiring 5 times as many tokens than others. Previous work has shown that tokenization in multilingual models is usually biased towards high-resourced languages in the pretraining data (Ács, 2019; Rust et al., 2021); we observe that this is not always the case, but it could also be dependent on linguistic features or properties of language scripts.

**RQ2 (cost): do non-uniform tokenization rates lead to LM API cost disparity for speakers of different languages?** LM APIs like ChatGPT are available worldwide and have been widely claimed to have multilingual capabilities (Kasai et al., 2023; Lai et al., 2023).[6] We show that disparate fragmentation rates across languages lead to significantly high usage costs for less represented languages, and we argue for a more equitable API pricing system.

**RQ3 (model utility): do non-uniform tokenization rates affect the models' utility?** LMs have exhibited in-context learning capabilities, performing new tasks with few demonstrations as input (without parameter finetuning). This is highly desirable in any LM API as it avoids computational, annotation (and financial) costs. We show that high fragmentation rate of a language negatively affects the in-context learning performance in that language, resulting in reduced model utility.

**RQ4 (socio-economic aspects): what are the socio-economic implications of the API's cross-lingual cost and performance disparity?** Our analysis shows evidence that not only are LMs more expensive for certain languages, they are also less effective for them. To highlight the implications of these findings, we correlate those measurements with the socio-economic indicators of language speakers as a proxy for affordability of the APIs. This analysis indicates that *users who likely cannot afford high API costs are charged more for poorer service*, hindering uniform accessibility.

---

[6] https://help.openai.com/en/articles/6742369-how-do-i-use-the-openai-api-in-different-languages

## 3 Experimental Setup

### 3.1 Models

Throughout this work, we focus on two LMs: ChatGPT (Ouyang et al., 2022; Brown et al., 2020) (gpt-3.5-turbo) and BLOOMZ (Muennighoff et al., 2022). Both of these models are trained and advertised as general-purpose models capable of following instructions and performing a wide range of tasks (Qin et al., 2023; Zhu et al., 2023; Ahuja et al., 2023; Huang et al., 2023).

ChatGPT (Ouyang et al., 2022) is a closed model only accessible through an API (with a premium tier) provided by OpenAI. Studies report that it supports as many as 90 languages (Ahuja et al., 2023). ChatGPT can handle a maximum sequence length of 4096 tokens (including both the prompt and generated tokens).

BLOOMZ (Muennighoff et al., 2022) is an open-source multilingual model trained on 46 natural languages and 13 programming languages. While training its tokenizer, sentences from different languages were sampled according to a multinomial distribution (Conneau et al., 2020), thereby increasing the number of tokens associated with low-resource languages. The best-performing version of this model has 175B parameters and is not feasible to be loaded on our academic servers; hence we rely on a free API of BLOOMZ hosted by Huggingface.[7] Although BLOOMZ was trained with ALiBi positional embeddings (Press et al., 2022) which allows the model to extrapolate to any length sequences during inference, the Huggingface API has a context limit of 1000 tokens.

### 3.2 Tasks and Datasets

To answer RQ1—whether the same information is conveyed with similar numbers of tokens in different languages—we use a validation set of FLORES-200 (Goyal et al., 2022), a multilingual parallel corpus containing examples in over 200 languages.[8] We tokenize each sentence in the FLORES-200 subset with ChatGPT's tokenizer[9] and compute the average number of tokens per sentence for each language. Using parallel data controls for the same information across languages. We consider that

---

[7] https://huggingface.co/docs/api-inference/quicktour.

[8] We also experimented with WMT 2021 data (Akhbardeh et al., 2021) and found similar results. Note that the WMT data are focused on European languages.

[9] ChatGPT's tokenizer https://github.com/openai/tiktoken

language A is more efficiently tokenized than language B if it uses fewer tokens per sentence on average. While previous studies have computed fragmentation rates with fertility (Ács, 2019), we instead define it as the average number of tokens in a sequence for two reasons. First, our goal is to compare LLM API costs across languages that charge users based on the number of tokens. To control for content, we use a parallel corpus for this analysis. Second, many languages we analyze are understudied and do not have word tokenizers available which are required to compute fertility.

For RQ2 and RQ3, to clearly highlight the cost and utility disparities, we evaluate the models on NLP tasks that involve long-form texts either at input or output. We evaluate the models on diverse, challenging natural language generation and classification tasks on the following benchmarks:

**Classification**  We evaluate on (1) XNLI (Conneau et al., 2018): a cross-lingual inference benchmark comprising of 11 typologically diverse languages. It involves two sub-tasks, passage selection and minimum answer span (Gold-P). We focus on the latter task in our experiments. (2) XFACT (Gupta and Srikumar, 2021): a multilingual fact verification dataset of naturally existing real-world claims covering 25 languages.

**Span Prediction**  We use XQUAD (Artetxe et al., 2019): a crosslingual question-answering dataset where each example consists of a paragraph, a question, and the answer as a span in the paragraph.

**Generation**  We evaluate on (1) Cross Sum (Hasan et al., 2021a): a cross-lingual abstractive summarization dataset comprising 1.7 million article-summary samples in 1500+ language pairs, and, (2) XLSUM (Hasan et al., 2021b): a summarization dataset covering 44 diverse languages. It comprises news articles and summaries in the same language as the article.

### 3.3 Prompting Formulation

We evaluate both models in a $k$-shot in-context learning setup where we also provide task instructions. We experiment with $0 \leq k \leq X$, where $X$ is the maximum number of in-context examples that can be provided. Note that $X$ is not a fixed value, but is determined by the LM API's limit on the number of input tokens and the fragmentation rate of the language.

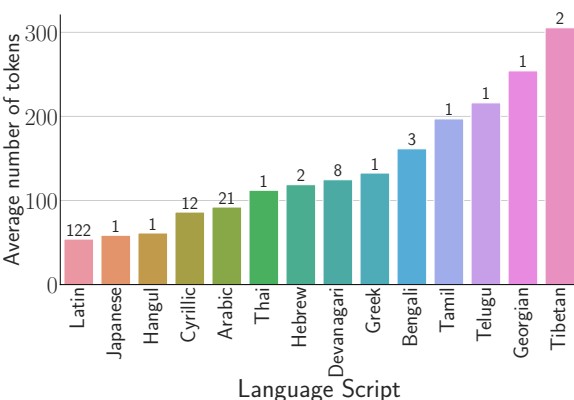

Figure 2: Average number of tokens by script after tokenizing the Flores dataset. The fragmentation rate is lower for Latin script languages and higher for other scripts. Number of languages per language group is indicated at the top of each bar.

For all tasks, we provide the instructions in English following Ahuja et al. (2023), who show that on several multilingual benchmarks, English instructions outperform the in-language prompts (see Table 2 in the Appendix for the prompting format for all tasks). For each task, we randomly sample at most 500 test examples for evaluation.

## 4 Results and Analysis

### 4.1 RQ1 (number of tokens): do all languages convey the same information with the same number of tokens?

In Figure 2 we show that Latin-script languages are represented with substantially fewer tokens compared to languages in other scripts. While Cyrillic and Japanese script languages come close to the Latin, languages with their own script (e.g., Telugu) require up to 5× more tokens to convey the same information. We hypothesize that this disparity is due to training data imbalance since ChatGPT's tokenizer was primarily trained on Latin-script languages, mainly English. The training details of ChatGPT are not available. However, we make a reasonable assumption that its training data has a similar proportion of languages as the publicly available large corpus CC100 (Wenzek et al., 2020). If we sort languages shown in Figure 2 based on their data size in CC100 (see Figure 14 in the Appendix), low-resourced languages of Latin script appear to be less fragmented compared to other mid-resourced languages of non-Latin scripts.

In Figure 15 in the Appendix, we present a similar analysis for BLOOMZ's tokenizer. We sort

the languages based on their size in the pretraining data (ROOTS corpus; Laurençon et al., 2023). We observe that languages with fewer resources generally have a higher average token length. Arabic is an outlier here as it appears to have more tokens than some other mid-resourced languages.

**What influences the non-uniformity of a tokenizer across languages?** From our analysis above, we identify two influential factors: (1) the proportion of the language in the pretraining data, and (2) inherent properties of the language and its writing script. While we see some correlation between pretraining data size and fragmentation rate in BLOOMZ , with ChatGPT it is quite different as higher-resourced non-Latin script languages still get excessively tokenized.

To disentangle the effects of factors (1) and (2) we train BBPE tokenizers on a variety of languages with diverse scripts with vocabulary sizes ranging from 5,000 to 50,000, while controlling for content and data size. Specifically, we train the tokenizers on parallel corpora and include one language per script. We then use these tokenizers to tokenize the text they were trained on, and compute the average number of tokens per sentence.

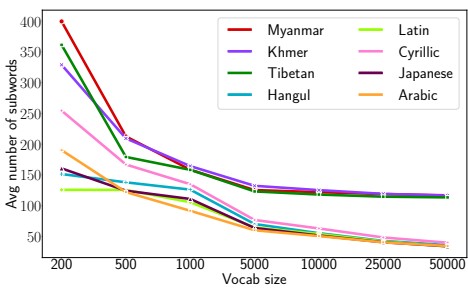

Figure 3: BBPE tokenizer trained on parallel text from different language scripts with varying vocabulary sizes. We display a larger version with 21 more scripts in Figure 19 in the Appendix.

As shown in Figure 3, even when controlling for the content, there is still a disparity in the tokenization rate at different vocabulary sizes. In particular, most scripts are very sensitive to small vocabulary sizes compared to Latin and Hangul scripts. We do not achieve uniform fragmentation rate across all language scripts even with large vocabulary sizes. We therefore conclude that uniformity of BBPE tokenizers across languages is not just determined by the proportion of text from language in the pretraining data but also by language/script properties.

## 4.2 RQ2 (cost): how do non-uniform tokenization rates affect LM API costs for different languages?

LM APIs charge users a fixed amount for a given number of input and generated tokens. Since the same information is expressed using different number of tokens in different languages, we aim to investigate the disparity in what users pay to use the API for different languages. From the results of our analysis in §4.1, we compute the estimated cost of API use per language as a function of the average sequence length derived in Figure 2. We report this on a subset of languages in Figure 16 in the Appendix and present a granular analysis of languages that share family and script in Figure 4.

Languages that are more heavily segmented have predictably higher costs of usage. Overall, we see that the API costs are biased towards (i.e., cheaper for) Indo-European and Latin script languages and against many non-Latin script languages. In most mid-resourced Indic languages with non-Latin scripts, we see close to a 5× increase in cost compared to English.

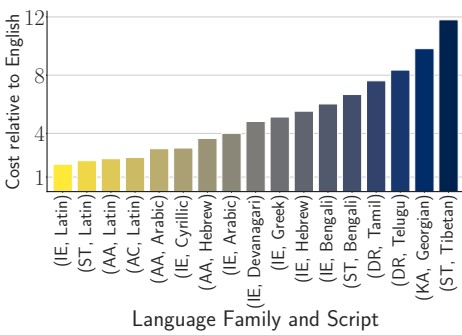

Figure 4: Estimated cost per language family/script, relative to English. The language families are abbreviated as follows: IE: Indo-European, ST: Sino-Tibetan, AC: Atlantic-Congo, AA: Afro-Asiatic, DR: Dravidian, KA: Kartvelian.

Next, we report the costs of running experiments relative to English. We report costs based on our zero-shot experiments across all tasks listed in §3.2. This is due to excessive tokenization in some languages for which we can only do zero-shot evaluations. For XLSUM, we show in Figure 5 that we spend up to 4× more for both prompting and generation in Telugu and Amharic. We observe similar findings in XFACT and CROSSUM, as displayed in Figure 11 in the Appendix.

While the majority of the commercial LMs are perhaps being optimized to perform well in many

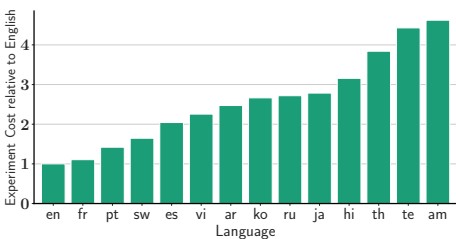

Figure 5: Average cost of prompt + generated tokens for XLSUM evaluations relative to English.

languages, we show that there is less focus on individual experiences of speakers of languages other than English. While LMs like ChatGPT might perform tasks in Telugu, for example, a user in Andhra Pradesh might pay 5× more than an English user in the US for an equivalent use of the model.

### 4.3 RQ3 – Model utility: do non-uniform tokenization rates affect the models' utility?

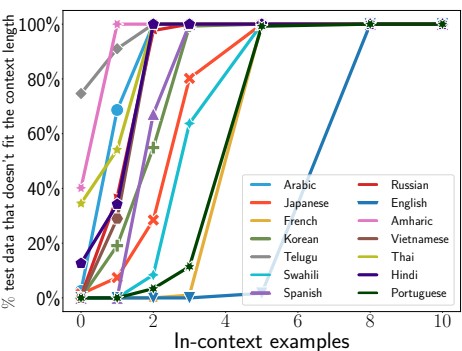

Figure 6: Percentage of test examples per language in XLSUM that do not successfully fit into the context length of ChatGPT. We can fit more few-shot examples in Latin script languages than in other languages.

LMs typically have an upper bound of the number of tokens they can handle, e.g., ChatGPT can process a maximum of 4,096 tokens. Hence, due to non-uniform fragmentation rates across languages, there is a disparity in the amount of information the models can process per language. In Figure 6 we plot the percentage of XLSUM test instances against the maximum number of in-context examples those instances can be accompanied with. For example, Telugu struggles to fit even one in-context example for the majority of the test set. Hence, the model can only do zero-shot prompting in this case.

To measure the impact of this issue on task performance, we evaluate ChatGPT and BLOOMZ with a $k$-shot learning setup on the 5 tasks on di-

verse languages as described in §3.2. Figure 7 shows ChatGPT's performance according to standard automatic metrics of all tasks. Note that the focus of this experiment is to illustrate the impact of tokenization in in-context learning settings. Therefore, we are interested not in the absolute value of the metrics or comparisons among languages but the relative improvement within the test sets of the same language as we increase the number of in-context examples. For all tasks and most languages, we see consistent performance improvements as we increase the number of in-context examples, from zero-shot to $k$ (even for $k = 1$). For many languages such as Telugu and Thai, due to their high fragmentation rates, we were unable to fit even one complete demonstration and hence, only report zero-shot results. Based on trends from other languages, we suspect that these languages could also have benefitted from more demonstrations. Hence, as a result of unfair tokenization, ChatGPT's utility is much lower for speakers of those languages compared to better represented languages like English.

Figure 8 reports the results of the same experiment for BLOOMZ. Across all tasks we find that adding in-context examples does not help. In fact, in some cases, there is a performance drop even with one in-context example. Upon manual inspection of the generated outputs from the one-shot experiments, the model has a tendency to copy spans from the in-context example, presenting that as output and thus not successfully utilize demonstrations. Our hypothesis here is that BLOOMZ is better optimized for zero-shot prompting and is not as suitable for in-context learning.

Due to the limited number of tokens that BLOOMZ's inference API accepts, some examples in some languages cannot fit the 1000 token context length when doing zero-shot prompting. We experienced this with the XLSUM dataset as we couldn't fully fit news articles for some languages. Understandably, some of these languages are not even present in its pretraining data, and hence we do not expect them to be tokenized efficiently. For these examples that do not fit the context length, we feed in truncated news articles into the model. We therefore evaluate the generations for the fraction of examples that fit context and ones that do not fit the context separately. Figure 9 shows the performance comparison when we use truncated summaries in the prompt and when we use the full articles. While the performance drop is expected,

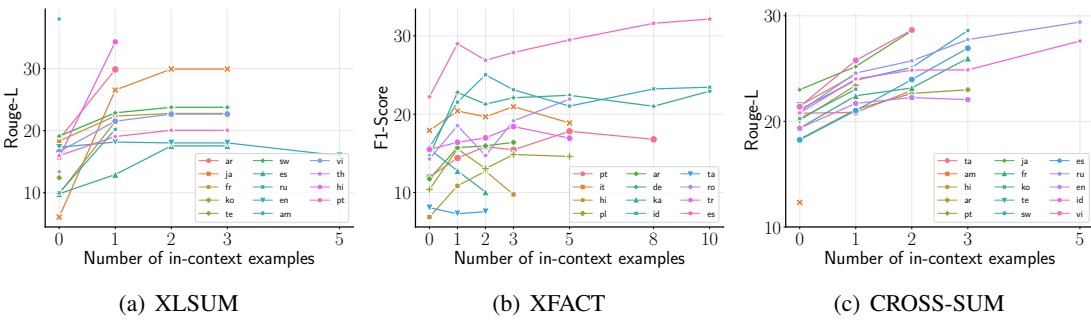

(a) XLSUM

(b) XFACT

(c) CROSS-SUM

Figure 7: Results from ChatGPT few-shot evaluations. In most tasks, we see an increase in performance as we increase the number of in-context examples.

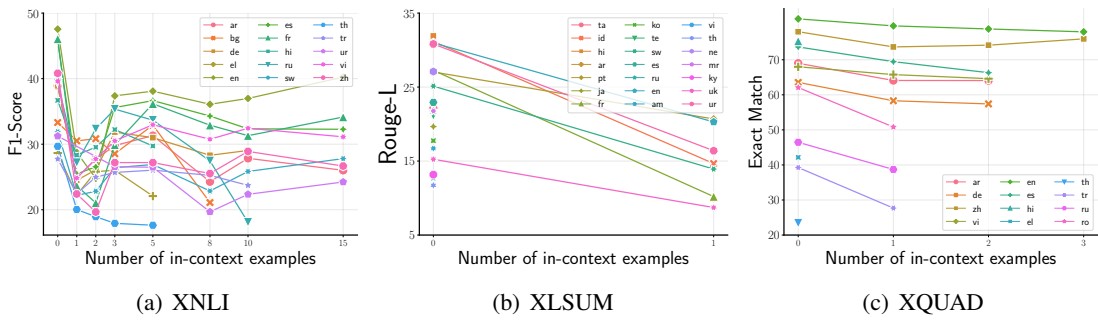

(a) XNLI

(b) XLSUM

(c) XQUAD

Figure 8: Results from BLOOMz few-shot evaluations. The BLOOMz model is clearly better at zero-shot prompting than few-shot.

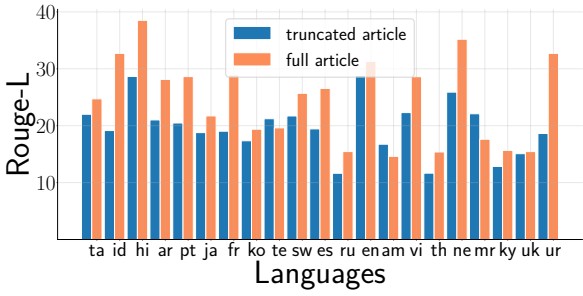

Figure 9: Zero-shot evaluation of BLOOMz on XL-SUM. Since we cannot fit the full article in the context length for some languages, we compare results on evaluating full articles vs. truncated articles.

our focus here is to highlight a consequence of differentiated tokenization in LMs.

### 4.4 RQ4 – Socio-economic aspects: what are the socio-economic implications of the API's cross-lingual cost and performance disparity?

In Figure 10, we plot the fragmentation rate per language against the Human Development Index in the country with the highest absolute number of speakers of that language. We find a strong negative correlation close to -0.5 showing that in

most cases, the lower the HDI index, the higher the fragmentation rate and vice versa. Evidently, the model's vocabulary is biased towards users of more developed countries.

| Task | Cost-HDI | | HDI-Utility | | Cost-Utility | |
|------|----------|---------|-------------|---------|--------------|---------|
| | Spearman | Pearson | Spearman | Pearson | Spearman | Pearson |
| XFACT | **–0.41 | **–0.60 | *0.34 | **0.38 | **–0.61 | **–0.55 |
| XLSUM | **–0.42 | **–0.43 | **–0.44 | **–0.57 | *–0.23 | *0.21 |
| CROSS SUM | **–0.41 | **–0.45 | *–0.18 | *0.24 | *0.27 | *–0.17 |

Table 1: Correlation between model utility, cost of API access and Human Development Index (HDI) for each task. We mark correlations with $p < 0.05$ with * and also mark correlations with $p < 0.0056$ (according to Bonferroni correction for multiple hypotheses) with **.

This bias is further validated by results shown in Table 1, where we mostly find negative correlations between pairs of each of the following variables: average financial cost of experiments, model utility (performance), and human development index of the country in which each language is spoken. We term this "double unfairness" as people from less economically developed countries are overcharged at a fixed rate per-token due to excessive tokenization, but often derive less utility from the model.

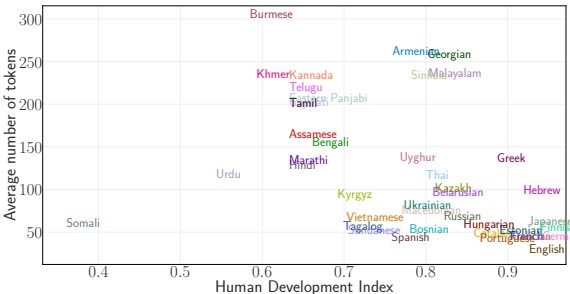

Figure 10: Fragmentation rate per language against the Human Development Index in the top country where the language is spoken.

## 5    What is the Way Forward?

**Transparency in API limitations**   While the NLP community is aware of many of the issues we point out in this work, LM APIs are advertised to a general audience. Much like the policy of adding limitations to research papers, LM API providers ought to be more transparent about the flaws and biases in their models, especially when describing their multilingual capabilities. Many users are not privy to the inner workings of these models and will be unknowingly charged higher prices if they use the model in their native languages.

**Rethinking the API pricing models**   Higher API costs for languages in underprivileged communities risks excluding many populations from using language technologies. A potential solution is to develop pricing policies based on languages/regions while also accounting for model performance on language-specific benchmarks. An alternative is to not charge by tokens at all. PaLM 2 API, for example, charges the users based on characters.[10] Hence, further analysis is needed to assess the fairness of character-based pricing. Huggingface also offers an inference service for their enterprise customers[11] relying on AWS instances and charging them at an hourly rate. Future work may compare this with a per-token rate we study in this work.

**Open-source models vs. paid APIs**   Given issues with paid APIs, the natural next question might be: should the API users move to open-source models or train their own? In fact, in our experiments, we find BLOOMZ, an open-source model, to perform better in the zero-shot setting than ChatGPT

---

[10]Prior work has shown evidence that even the number of characters used to express the same information in different languages is

[11]https://huggingface.co/pricing#endpoints

performs in the few-shot setting, in most cases. However, first, most open-source models are distributed under an academic license whereas most developers are interested in integrating these technologies into their products for commercial use, which may incur licensing costs. Second, barring licensing issues, LMs tend to be large and resource-intensive to train and deploy and require dedicated expensive hardware to run at a commercial scale, which again might not be possible for most developers and users, even exceeding the cost of using the APIs. Research on reducing such hardware requirements (Dettmers et al., 2022; Park et al., 2023) could increase accessibility. Still, this requires a considerable level of technical expertise from developers and users which might be infeasible.

**Technological improvements in LMs**   Several solutions proposed in recent work to improve language modeling performance can help alleviate the cost and utility issues we highlight. Tokenization is an active area of research and various solutions based on data balancing (Johnson et al., 2017; Conneau and Lample, 2019), optimal transport (Xu et al., 2021), fuzzy subwords (Provilkov et al., 2020), and many more (Chung et al., 2020; Tay et al., 2022) have been proposed. BLOOMZ, for instance, relies on data balancing to improve fragmentation rates across languages. Some works also focused on increasing the context lengths of language models (Bulatov et al., 2023; Press et al., 2022) which can help alleviate issues with utility by allowing more in-context examples as input.

## 6    Related Work

**Analyzing tokenization methods**   The impact of tokenization on model performance  (Ács, 2019; Rust et al., 2021; Zhang et al., 2022a; Klein and Tsarfaty, 2020; Bostrom and Durrett, 2020; Kamali et al., 2022), inference speed and memory usage of LMs in practical settings (Sun et al., 2023; Hofmann et al., 2022) has been widely studied. Ács (2019) observes that mBERT's vocabulary is largely dominated by Indo-European languages. Rust et al. (2021) find that monolingual LMs perform better than mBERT because some languages suffer from over-fragmentation. Zhang et al. (2022a) find that sentence-level MT models are not sensitive to language imbalance in their tokenizer training data. In contrast to prior work, our focus is on the cost and performance analysis of multilingual LM APIs across languages with regard

to over-fragmentation and in-context learning.

**Socio-economic impacts of language models**
Prior work show that unfairness in LMs is a consequence of many stages in the development pipeline (Cao and Daumé III, 2020; Talat et al., 2021). Efforts have tried to identify social biases in LM generations (Wolfe and Caliskan, 2021; Dev et al., 2022; Sheng et al., 2021; Chen et al., 2021; Hutchinson et al., 2020). Other works have surfaced the cultural and language disparity beyond and within multilingual LMs (Gururangan et al., 2022; Kreutzer et al., 2022; Virtanen et al., 2019). Talat et al. (2022) discuss challenges impacting bias evaluation in multilingual LMs. They examine power dynamics and consequences of training LMs emphasizing implications associated with advancement of such technologies. In this work, we study economic unfairness of LMs across different communities. Concurrent work (Petrov et al., 2023) analyses multilingual tokenizers focusing on financial cost, latency and context size. However, apart from cost, our analysis also covers model utility and socio-economic implications. Kasai et al. (2023) report unfair API costs as a result of tokenization differences between English and Japanese. We extend this to 21 more languages highlighting the pervasiveness of this issue.

## 7 Conclusion

By analyzing popular language model APIs on challenging multilingual benchmarks, we find that (a) API tokenizers disproportionately favor Latin scripted languages and over-fragment less represented languages and scripts, (b) the API pricing policy of charging based on the number of tokens is flawed and extremely unfair towards speakers of the over-fragmented languages, and (c) the API performs poorly on such languages compared to the less-fragmented counterparts. In the current NLP research landscape, where more and more industrial labs are building their own APIs, this is a concerning trend that may reduce the accessibility of these technologies to already marginalized communities. Hence, we encourage the vendors to be more transparent about their models' limitations and rethink their pricing policy.

## Ethics Statement

This work sheds light on the consequences of unfair tokenization to users of commercial LM APIs that speak languages with scripts less represented in the pretraining data. With the recent widespread use of commercial LMs, we believe that our work is crucial to ensuring that language technologies are accessible to diverse users irrespective of the languages they speak.

There are different factors that contribute to non-uniform tokenization across languages. Whilst our analysis touches on the size of pretraining data and language writing systems we suspect that there might be other factors not yet uncovered; we leave that for future work. The lack of access to OpenAI's training data prevents us from making solid claims about all the languages that ChatGPT is optimized for; however, their models have been advertised and shown to work well in many languages. More work on large multilingual models should include the release of (details of) training data to further enable this kind of research.

## Limitations

**Translationese** We conduct the analysis to answer RQ1 using a parallel corpus, FLORES-200 (Team et al., 2022), in order to control for the same information. This corpus consists of many examples that have been professionally translated. Prior studies have shown that translated texts in any language (referred to as translationese) may differ from original written text in many ways (Laviosa, 2002). These may have caused the information conveyed in different languages to not be exactly the same. We do not have a way to measure these differences. However, we expect them not to be so large as to meaningfully affect the trend of fragmentation rates.

**Language statistics of ChatGPT training data** ChatGPT is a closed model developed by OpenAI who have not released the training details of the model including any information of the languages it supports.[12] Hence, we cannot ascertain the actual statistics of all the languages in their training data. We use CC100 (Wenzek et al., 2020), a large multilingual corpus, to estimate these statistics.

**Reproducibility** One limitation with testing closed LMs is lack of reproducubility particularly

---

[12]The only official information they provide about Chat-GPT's multilingual support is here: `https://help.openai.com/en/articles/6742369-how-do-i-use-the-openai-api-in-different-languages` Prior studies have speculated that ChatGPT was trained on at least 90 languages (Ahuja et al., 2023).

because the model weights are typically updated continually. However, this only affects the downstream evaluations as our cost analysis is reproducible, since the tokenizers we evaluate are open-source.

## Acknowledgements

We thank the members of the Tsvetshop and Noah's ARK labs at the University of Washington for the valuable discussions and useful feedback. We thank Lucille Njoo for help with our illustrations. We also thank the reviewers and area chair for their valuable feedback. During the course of this study, S.K. was supported by Google Ph.D. Fellowship. We also gratefully acknowledge support from NSF CAREER Grant No. IIS2142739, the Alfred P. Sloan Foundation Fellowship, and NSF grants No. IIS2125201, IIS2203097, and IIS2113530. Any opinions, findings and conclusions or recommendations expressed in this material are those of the authors and do not necessarily state or reflect those of the United States Government or any agency thereof.

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

# A Prompt template

In Table 2, we provide the exact prompts we use for each respective task in our experiments.

## B Cost analysis

In Figure 11 we present the experimental cost relative to English and Spanish for Crosssum and XFACT respectively. Figure 16 shows the estimated cost of GPT3.5 API access for all languages in CC100 relative to English .

## C Extra analysis of fragmentation rate

In Figure 12 and Figure 13 we present the fragmentation rate across language families and scripts for both GPT3.5 and BLOOM respectively. Figure 17 shows the fragmentation rate for all languages in FLORES grouped by language script.

## D Fragmentation rate vs pretraining data size

In Figure 14 we sort languages based on their size in CC100 corpus (Wenzek et al., 2020) and plot their fragmentation rate with GPT3.5 tokenizer. Figure 15 shows the same statistics for BLOOM's tokenizer based on language pretraining data size in (ROOTS corpus; Laurençon et al., 2023).

## E Pretrained tokenizers on more languages

Figure 19 shows fragmentation rate across language scripts, when we train a BBPE tokenizer trained on parallel text in 30 languages.

## F Fragmentation rate vs HDI

Figure 18 shows GPT3.5's fragmentation rate per language against the Human Development Index of the country with the largest amount of speakers of that language. We add more languages and countries here compared to the figure in the main paper.

| Task | Prompt Template |
|---|---|
| XLSUM | Write a short summary sentence of the following text in {language} Article: { article} Summary: |
| XQUAD | Context: context Question: question Answer: Template |
| XNLI | {Premise} Question : {hypothesis} True, False, or Neither? Answer: |
| CROSSUM | Write a short summary sentence of the following text in English. Article: { article} Summary: |
| XFACT | Tell me whether the following claim is {label 1 } or {label 2 } or {label 3 } ... given evidence {evidence 1 }, {evidence 2 }, {evidence 3 } |

Table 2: Prompt template used for each dataset.

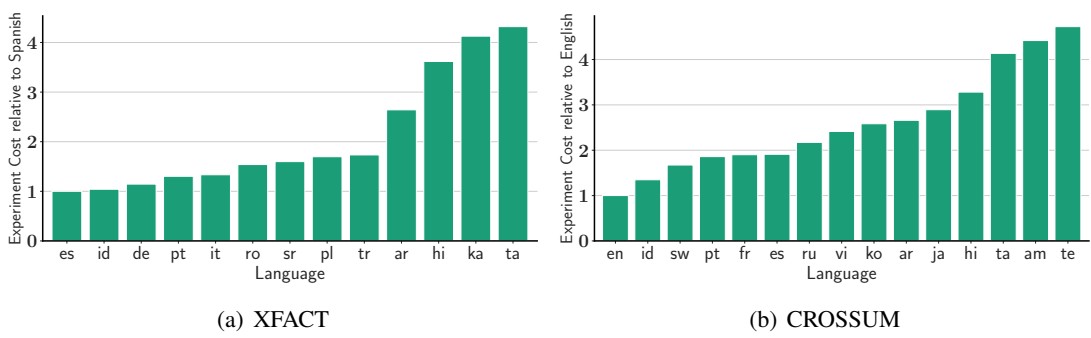

(a) XFACT

(b) CROSSUM

Figure 11: Relative cost of prompt + generated tokens for XFACT and CROSS-SUM evaluations.

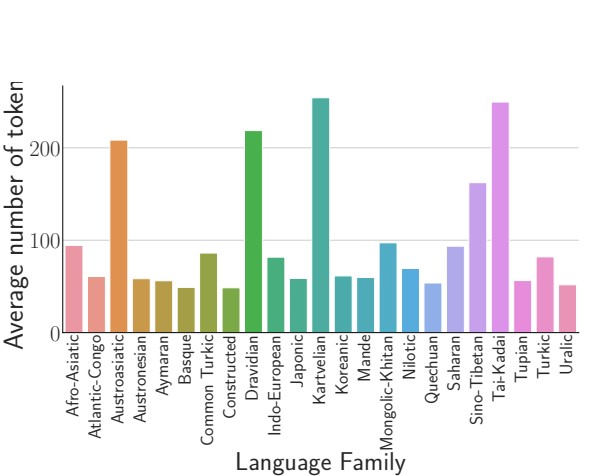

Figure 12: Average number of tokens per language family after tokenizing Flores dataset with GPT3.5 tokenizer. The fragmentation rate is lower for Latin script languages and higher for other scripts.

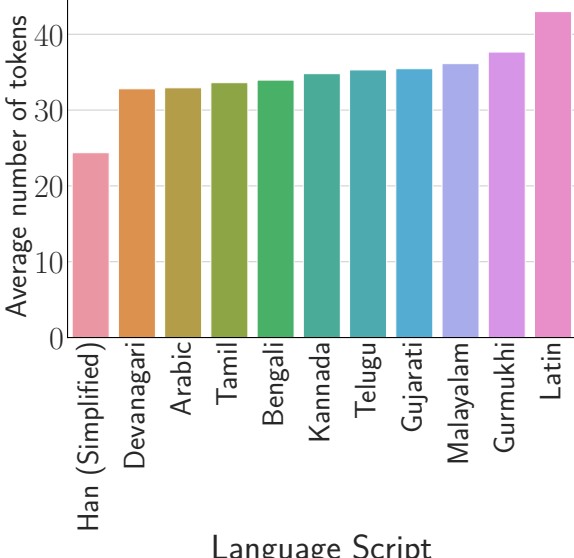

Figure 13: Average number of tokens per language script after tokenizing Flores dataset with BLOOM tokenizer. The fragmentation rate is higher on average for Latin script languages. This is because majority of the low-resourced languages are latin-script and have higher fragmentation rate.

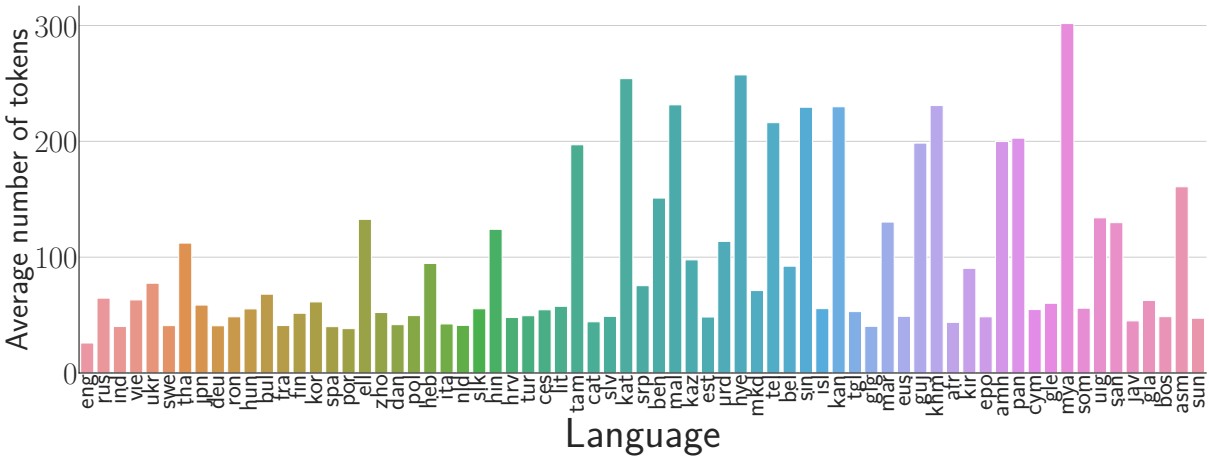

Figure 14: Average number of tokens per language after tokenizing FLORES with GPT3.5 tokenizer. Languages are arranged in descending order based on the size of pretraining data in Commoncrawl.

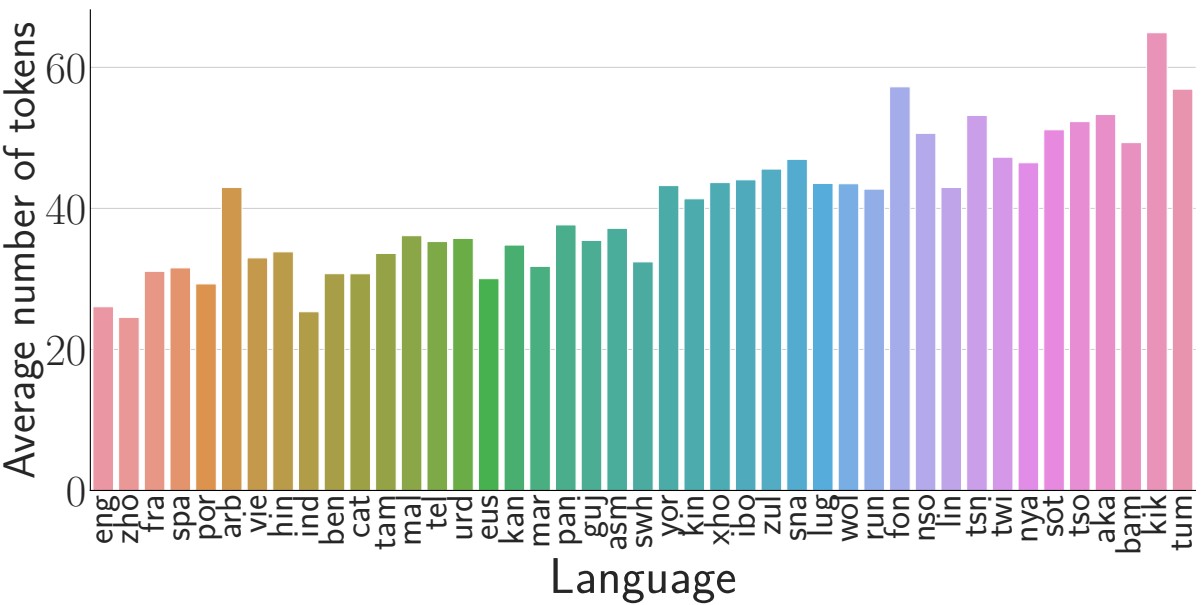

Figure 15: Average number of tokens per language after tokenizing FLORES with BLOOM tokenizer. Languages are arranged in descending order based on the size of pretraining data in the ROOTS corpus.

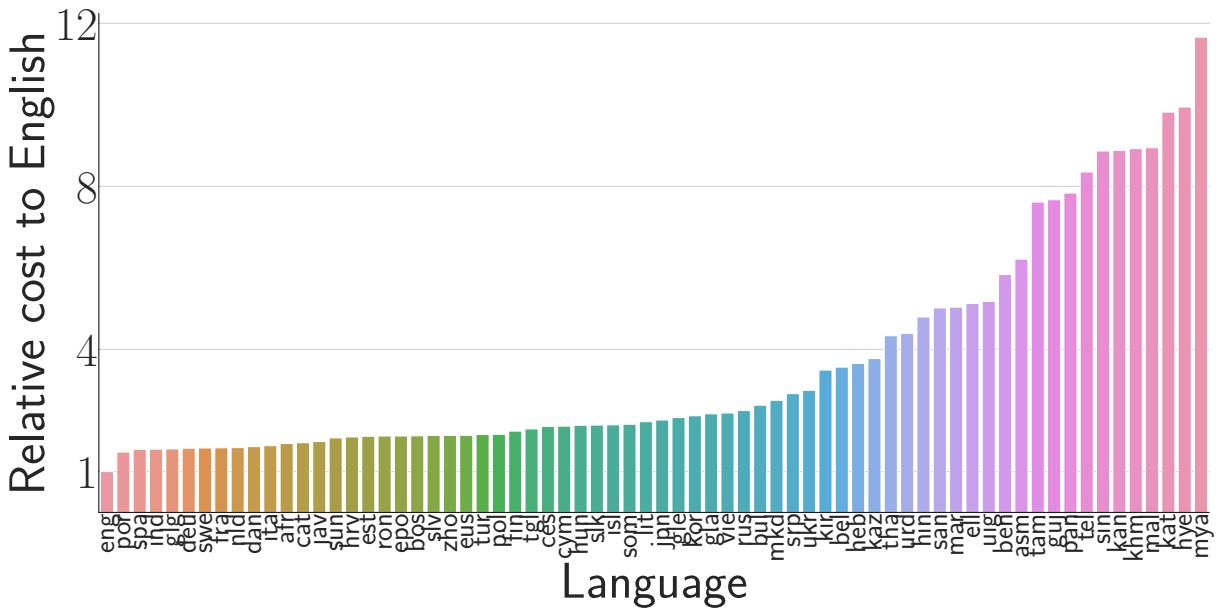

Figure 16: Estimated cost of GPT3.5 API access relative to English.

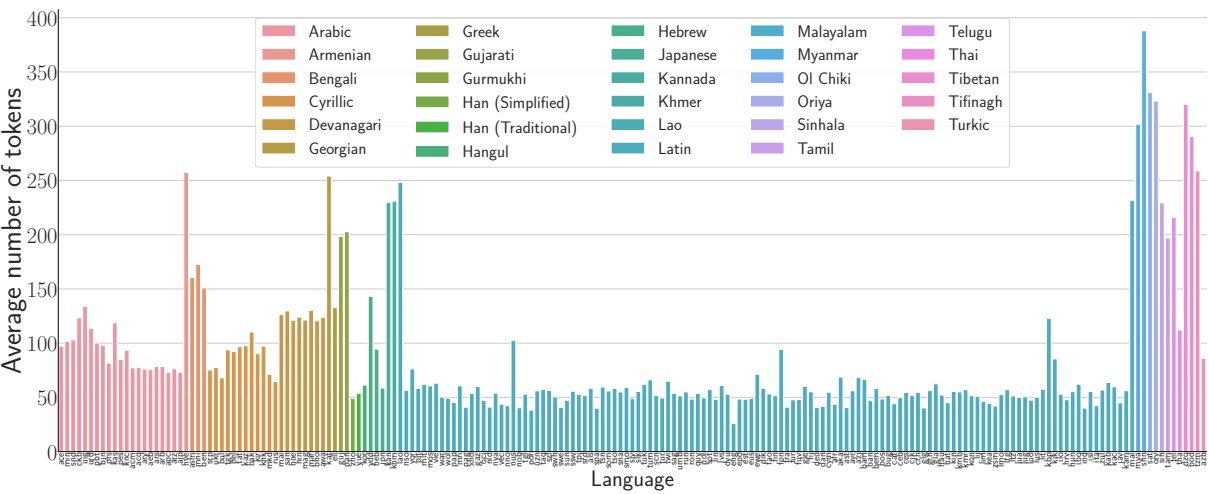

Figure 17: Average number of tokens by script after tokenizing all languages in the Flores dataset with GPT3.5 tokenizer.

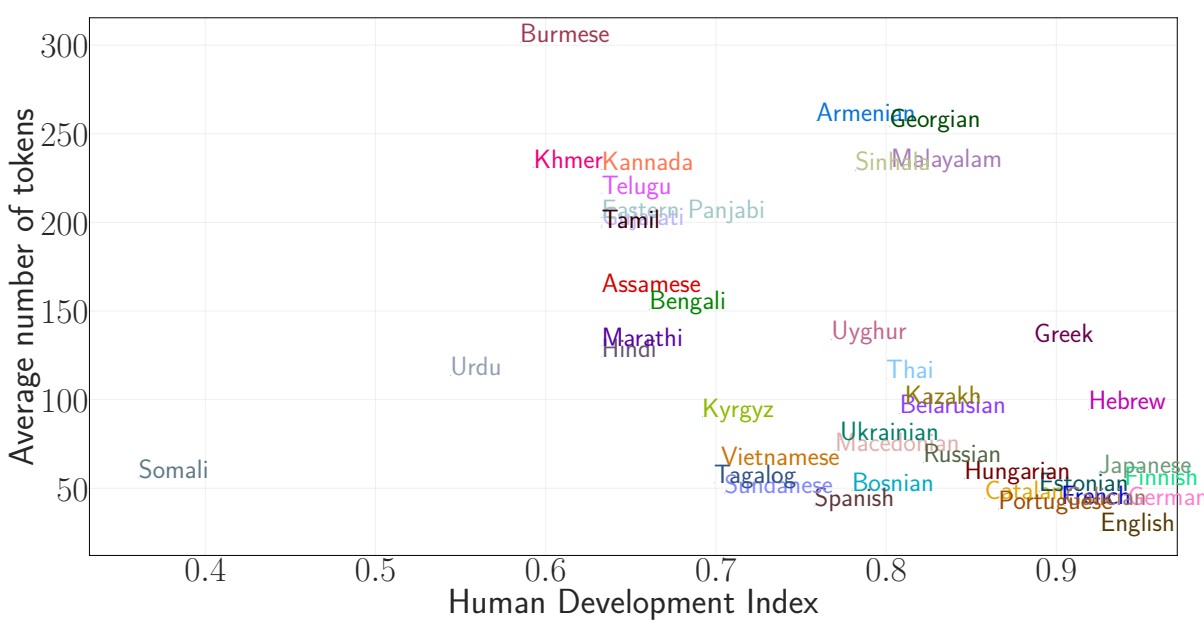

Figure 18: Fragmentation rate per language against the Human Development Index in the country with the largest amount of speakers of that language.

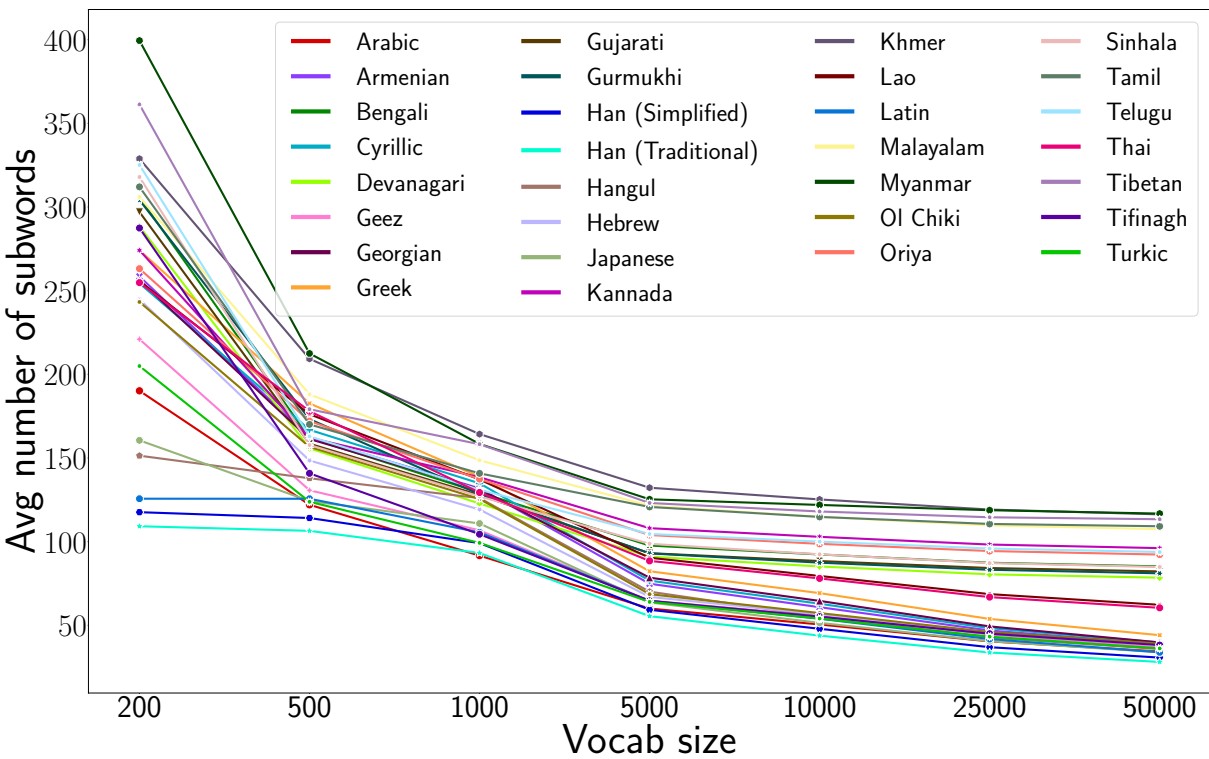

Figure 19: BBPE tokenizer trained on parallel text from 30 language scripts with varying vocabulary sizes. It is impossible to achieve uniform fragmnatation rate even when we have equal pretraining data sizes across all language scripts.