# OpenReview forum: "Do All Languages Cost the Same? Tokenization in the Era of Commercial Language Models"
_EMNLP/2023/Conference — EMNLP 2023 Main_

### Official Review · Reviewer_8QsM · 2023-07-21

**Soundness:** 2

**Excitement:**

3: Ambivalent: It has merits (e.g., it reports state-of-the-art results, the idea is nice), but there are key weaknesses (e.g., it describes incremental work), and it can significantly benefit from another round of revision. However, I won't object to accepting it if my co-reviewers champion it.

**Missing References:**


*
Claims like "There are hundreds of distinct writing systems in the world." should be backed up by a reference, e.g.

Daniels, Peter T. & William Bright (eds.). (1996) The world's writing systems. Oxford: Oxford University Press.

*
Correlations between computational language resources and the economic power of speakers has been done before, with somewhat better theoretical foundation, see e.g., pp 107-132 of

Hammarström, Harald. (2009) Unsupervised Learning of Morphology and the Languages of the World. Chalmers University of Technology and University of Gothenburg doctoral dissertation.


**Paper Topic And Main Contributions:**

The paper provides a series of measurements relating to the number of tokens and utility per token across different languages/scripts in commercialized LLM web APIs. These measurements are novel and interesting but conceptual-theoretical foundation of the paper is weak and the author(s) do not seem to know the languages they are studying so the discussion is suboptimal. Some key parts do not have a satisfactory description and some crucial experiments are poorly conducted.


**Reasons To Accept:**

A series of measurements relating to the number of tokens and utility per token across different languages/scripts in
commercialized LLM web APIs.

**Reasons To Reject:**

Conceptual-theoretical foundation of the paper is weak and the author(s) do not seem to know the languages they are studying
so the discussion is suboptimal. Some key parts do not have a satisfactory description and some crucial experiements are poorly conducted.


Major:
* The author(s) use value judgements ("fair", "unfair" etc) with the
tacit assumption that fair is that each language should have the same
price. They provide no philosophical or economical grounding for the
underlying assumption. But the token ratios depend on the amount of
training data and companies in question operate in a supply-demand
economy so it's natural that languages for which data is in less supply
is more expensive and that English is cheap.

* "Second, many languages we analyze are understudied and do not have word tokenizers 242
available which are required to compute fertility."

None of the languages analyzed is understudied in the relevant sense (i.e., wrt to tokenization). See, e.g., glottolog.org for descriptions of the phonology and writing systems of all. Nearly all have word tokenizers (only a few, e.g., Burmese is problematic in this domain).

* The results re RQ1 do not feature 101 or 200 languages either in the text (well, one grouped figure is there) or the appendix. Yet the authors assert correlations of various kinds where the remaining 150-ish languages are highly relevant. (To exlcude them in the presentation or analsysi is what is really unfair...!)

* The crucial Fig 2, 14-15 leave some things to desired. First, I want the
full list of 200 languages and the 13 programming languages to be included
in the BLOOMZ case (they are relevant because the tokenization rate of one
language crucially depends on what other languages there are in the training
data). Second, informal rank correlations are asserted whereas it seems
obvious a Pearson correlation is expected (maybe with the log of the data
size). Do formal Spearman or Pearson test if a correlation is asserted or
don't assert one. Third, the crucial observation "low-resourced languages
of Latin script appear to be less fragmented compared to other mid-resourced
languages of non-Latin scripts." is problematic since is it not said where
the border between mid and low resources languages go and why you are
compared low-resource latin with mid-resource non-latin rather that
mid to mid or low to low). Fig 14-15 (not Fig 2) is what should be in the
paper rather than the appendix because the RQ1 asks about languages *not*
scripts.

* Re the BLOOMZ and ChatGPT tokenizers. The referenced desc of the ChatGPT
tokenizer says it's just a BPE so the training setup is known, just not
the input data. The authors never explain what the Bloomz tokenizer does
but it seems it's doing something more than just (B)BPE because of the
much smaller discrepancies in tok rate, not quite expected from the disparate
input ROOTS corpus. Do explain.

* The crucial experiment reflected in Fig 3, 18 is completely
inadequate.  The comparison is *per sentence* not acknowledging that
sentence length can differ across languages, so what you might have
found is not diff tokenization rates but different sentence
length. (NB: Not all scripts indicate sentence boundaries but the
subset of languages studied appear to have such boundaries imposed in
the data as used). Presumably the vocabulary size is the vocabulary of
the BBPE tokenizer but different amount of bytes in different
languages are required to yield the same size vocabulary. The authors
say content is controlled for, ok, but HOW? Fig 18 has one language
per script, ok, but which language in each case? Finally it would be
nice to know how many sentences are required to reach a vocabulary of
50000?

* An important limitation is that the Flores corpus is translations ***from English*** so one can expect the other languages to be a little longer than the English original.

* "In particular, most scripts are very sensitive to small vocabulary sizes compared to Latin and Hangul scripts"

Not quite, in Fig 18 the logosyllabic Chinese are before Hangul. Anyway, it's not clear why this is relevant because the vocabulary sizes are probably very big in the Bloomz and ChatGPT tokenizers.

* The authors related the language-wide differences with HDI which is a composite of life expectancy, economic power etc. But the interpretations only relate to the economic power of the speakers ("afford", "higher cost" etc) and not the other factors in HDI. So the authors should have correlated directly with GDP and rather the entire purchasing power of the language not the average pp of a speaker. The data use is also substandard as only the "top" (in which sense?) country is selected, whereas speaker numbers across all countries are available in, e.g., Ethnologue.





Minor:
* "Previous work has shown that tokenization in multilingual models
is usually biased towards high-resourced languages in the pretraining
data (Ács, 2019; Rust et al., 2021);"

It follows theoretically from BBPE that frequent sequences in the
pretraining data get chunked more so if the pretraining data is
dominated by some language then it follows that the tokenization will
be "biased" towards that language. There is no need to invoke an
empirical study unless one is interested in the specific numbers.

* We use a subset of FLORES-200 (Goyal et al., 2022), a multilingual parallel corpus containing examples in over 200 languages.

What subset? The ref to Goyal et al. 2022 is for Flores-101 and not Flores-200. The difference is significant because the translation quality
for the addional 99 language will be quite different.

* The authors confuse "language", "script" and "script as used in the orthography for a language" throughout. Do fix. It's often ok to have a shorthand but not in this case where the relevant differences are explored.

* The ISO 639-3 code zho denotes a set of languages. Probably only cmn is intended.

* Give the size of the FLORES corpus (3000 sentences/language)

* It might be useful to look at pairs of languages that differ minimally
except for the script (e.g, croatian/serbian, uzbek-uigur at different times, hindi-urdu etc)



**Reproducibility:**

4: Could mostly reproduce the results, but there may be some variation because of sample variance or minor variations in their interpretation of the protocol or method.

**Reviewer Confidence:**

3: Pretty sure, but there's a chance I missed something. Although I have a good feel for this area in general, I did not carefully check the paper's details, e.g., the math, experimental design, or novelty.

**Typos Grammar Style And Presentation Improvements:**

Sec 2 includes the answers to the RQs, duplicating this info from the answers that follow in the results checking.

Figure 12 is orphaned (not referred to and whose relevant is not explained. Also it has errors like Turkic vs Common Turkic --- they are the same family, CT is the main branch of Turkic)

---

> ### Author Rebuttal · Authors · 2023-08-28
>
> We thank the Reviewer 8QsM for taking the time to review our work and offering valuable comments. We acknowledge their suggestions and will incorporate them.
>
> **Addressing your concerns** :
>
> **Conceptual-theoretical foundation of the paper**: Our scientific contribution in this work is methodological:  we provide a general methodology to evaluate the fairness and utility of language model tokenizers across linguistic and economic axes. Our methods rely on theoretically grounded socio-economic indicators and are linguistically informed. We believe such general methods are particularly relevant now, given the recent commercialization of large language models. Our methodology can be applied across standard languages, dialects and model architectures, and can inform future development of more equitable text tokenizers.
>
> **Fairness in LM API pricing**: In the paper, we highlight that the LLMs we study are advertised as capable of performing tasks in several languages. There are many factors that contribute to performance and the overall satisfaction users derive from these APIs. Our goal is to shed light on an often neglected factor, tokenization. Of course, we believe that token-ratios would often correlate with amount of pre-training data, we however believe that it is largely misleading for LLM companies to extol the multilingual capabilities of their models without considering the socio-economic impacts on users that speak lower-resourced languages.
>
> **Computing tokenizer fertility**:  For the purpose of our analysis, we define fragmentation rate as the average number of tokens after tokenizing a corpus. While fertility reflects the average number of subwords that correspond to a single word, we are instead interested in the total cost of tokenizing the same content, but in different languages. Since LLM companies charge users per token, we believe that this approach enables a fair comparison across languages, and is most relevant to our research question
>
> **Description of ChatgGPT and BLOOM tokenizers**: In line 297 we state that we hypothesize that disparity in ChatGPT’s tokenization could be attributed to training data imbalance. Given that ChatGPT is a closed API, we cannot ascertain the training setup. We only infer from previous studies that speculate that ChatGPT was trained on 90 languages. Upon acceptance, we will explicitly make this clear in the description of the tokenizer.  The BLOOMz tokenizer was trained on a corpus in which sentences from different languages are sampled according to a multinomial distribution, thereby increasing the number of tokens associated with low-resource languages. We will include this detailed description in the final version upon acceptance.
>
> **Results from analyzing all 200 languages in FLORES**: We conducted our analysis on all languages in the FLORES 200 dataset and decided to present an aggregated plot in the main paper due to space constraints. However our findings still hold across all languages in the FLORES 200 dataset.  We generally see that non-Latin-script languages have higher fragmentation rates irrespective of their level of resourcedness (we categorize resourcedness based on the FLORES 200 paper). Given additional space upon acceptance, we will include these charts in the main paper.
> Regarding analyzing BLOOM’s tokenizer on all languages, we see the same trends as above, except the fact that BLOOM has lower fragmentation rate on Arabic and some Dravidian languages present in its pre-training data. We will include these findings in the final version.
>
> **Controlled experiment reflected in Fig 3, 18**: When building BPE tokenizers, different languages have different allocations in the vocabulary and this has been largely attributed to pretraining data size and not inherent properties of the language. Our goal with this experiment is to actually see what you get when you have parallel data in all languages, this is our approach to control content. Of course we see that different amounts of bytes are required to yield the same size of vocabulary in different languages.  We highlight this to prove that data imbalance isn’t the sole cause of disparate fragmentation rates across languages. In the final version, we will include the actual language selected per script for this analysis.
>
> **HDI correlation**s: We made the decision to use HDI as a proxy for socio-economic indicators, following prior work [1]. Notably, GDP already has a high correlation with HDI and we thus expect the same correlations with fragmentation rates to hold for GDP as well. We will include this analysis in the final version.
> [1] Incorporating Dialectal Variability for Socially Equitable Language Identification](https://aclanthology.org/P17-2009) (Jurgens et al., ACL 2017)
>
> **Subset of FLORES used for RQ1 experiments** : We used the validation set of FLORES 200, and will make this explicitly clear in the final version of the paper.
>
> **Vocabulary sizes of ChatGPT and Bloomz**: The vocab sizes of BLOOM and ChatGPT are significantly different. ChatGPT has a vocabulary size of 50257 compared to BLOOM that has 250080.

---

### Official Review · Reviewer_JDu5 · 2023-08-01

**Soundness:** 4

**Excitement:**

4: Strong: This paper deepens the understanding of some phenomenon or lowers the barriers to an existing research direction.

**Paper Topic And Main Contributions:**

Paper offers an extensive evaluation of disadvantages related to large language models in two external services. The evaluated services are the popular open AI model chatgpt and an open source alternative bloomz over huggingface api. The impacts are evaluated from the point of view of tokenisation with bbpe and overall use case. The evaluations highlight a very important aspect of inequality between languages written in latin alphabet and others. The tests are performed on typologically varied range of languages.

**Questions For The Authors:**

A. I think one of the problems wrt testing closed source LLMs via web apis is total lack of reproducibility, but perhaps this is explained in the article and needs more underlining?
B. Have you considered relating this research to morphological complexity and agglutinativity of the languages? E.g. maybe highly agglutinative as in almost concatenative languages still have more cost-friendly tokenisations?

**Reasons To Accept:**

* The points that are brought up in this article are highly important and relevnat tothe context of this conference.
* extents of testing in terms of language range are good and will provide good basis for discussion

**Reasons To Reject:**

* I am a bit torn on the scientific concept of an article based on just testing two off-the-shelf systems and providing mainly non-linguistic cost evaluation, however, I think this article is important and unique enough to warrant inclusion, especially as the focus of EMNLP does tend towards this direction anyways

**Reproducibility:**

2: Would be hard pressed to reproduce the results. The contribution depends on data that are simply not available outside the author's institution or consortium; not enough details are provided.

**Reviewer Confidence:**

2: Willing to defend my evaluation, but it is fairly likely that I missed some details, didn't understand some central points, or can't be sure about the novelty of the work.

**Typos Grammar Style And Presentation Improvements:**

The article reads well and is easy to follow.

Some formulations are a bit hard to follow, e.g.:
* on L240 "Cyrillic and Japanese scripts[...]languages with their own script e.g. Telugu", I guess Japanese is not strictly language with its own script but I am not sure if this expression is actually relevant to point being made here.

---

> ### Author Rebuttal · Authors · 2023-08-28
>
> We thank Reviewer JDu5 for their time, thoughtful feedback and for noting that our work is important and relevant to the wide NLP community.
>
> **Addressing your concerns**:
>
> **Scientific concept of the Paper** : Our scientific contribution in this work is that we provide a general methodology to evaluate the fairness and utility of language model tokenizers across linguistic and economic axes. We believe this is particularly relevant given the recent commercialization of large language models, and because text tokenizers are used in every such model. Our methodology can therefore be applied across standard languages, dialects and model architectures. Our methods also rely on theoretically grounded socio-economic and linguistic indicators.  Further, the GPT series of models are one of the most widely user-adopted applications to date, and therefore we believe a study focusing solely on them has great scientific and practical value for the research community.
>
> **Lack of Reproducibility**:  Reproducibility is definitely a limitation with testing closed LLMs. Importantly, the tokenizers we evaluate are open-source, and our fragmentation analysis is therefore reproducible. We will release the code to reproduce our analyses. With regard to downstream evaluations, we will ensure to explicitly include this in the limitation section of the paper.
>
> **Morphological Complexity**: Thank you for your suggestion, yes we definitely believe that morphological complexity of a language plays a role in tokenization quality. In fact,  we compared fragmentation rates across language families with different morphological complexities. Uralic languages like Finnish and Hungarian for instance have slightly higher fragmentation rates than other Latin languages that might have less resources available. We will include plots from this analysis in the final paper.
>
> **Summary**: we greatly appreciate your time and the careful review! We will incorporate discussions about scientific contributions, reproducibility, and morphological complexity-related analyses in the final revision.

---

### Official Review · Reviewer_YUzp · 2023-08-04

**Soundness:** 5

**Excitement:**

4: Strong: This paper deepens the understanding of some phenomenon or lowers the barriers to an existing research direction.

**Missing References:**

Missing references on LLM as APIs, e.g. https://arxiv.org/abs/2201.03514

**Paper Topic And Main Contributions:**

This paper investigates the disparate pricing among different languages on public LLMs APIs. They present an exhaustive methodology which yielded the following interesting findings:
- Tokenizers of popular LLMs over-fragment texts in certain language scripts
- They quantify the API cost disparity that this issue causes
- They show that the tokenizer pretrainind data is not the sole culprits but also show that this stems from language properties and the way the language is represented in unicode
- They show that such languages with over-fragmentation derive lower model utility from LLM APIs
- They show that languages costing more come from countries and regions with lower Human Development Index (HDI), further exacerbating the economic divide.
- Present discussions and suggests ways to remedy the situation.

Overall, their findings present interesting and critical insights given the prevalence of public LLM APIs.

**Questions For The Authors:**

- Why did you not use code for BloomZ to circumvent the API restrictions?
- Why did you not evaluate Anthropic's Claude, Cohere, and GPT4?
- Your results in figure 2 - are they generalizable given the small sample size of the scripts with the highest number of average tokens?

**Reasons To Accept:**

- Exhaustive methodology along different axis - monolingual & multilingual models, model performance and socioeconomic implications.
- Important findings on language properties and model utility based on tokenizer over-fragmentation of such languages.
- Showcase the socioeconomic insidious implications of this.
- Insightful discussions and recommendations for LLM practitioners and API builders
- Evaluations on classification, span prediction and text generation

**Reasons To Reject:**

- Only perform evaluation on ChatGPT and BloomZ. Missing Anthropic and Cohere.

**Reproducibility:**

5: Could easily reproduce the results.

**Reviewer Confidence:**

5: Positive that my evaluation is correct. I read the paper very carefully and I am very familiar with related work.

---

> ### Author Rebuttal · Authors · 2023-08-28
>
> We thank Reviewer YUzp for taking the time to review our paper, and noting that our work is critical and insightful in analyzing socio-economic implication of tokenization in LLMS. We are glad that they find our work particularly relevant to the wide NLP community.
>
> **Addressing your concerns**:
>
> **Circumventing BLOOMz API restrictions with code**: BloomZ is a 176B parameter model. Due to compute limitations in academic settings, we couldn’t load the model on our servers. We believe that is also the case with many researchers that cannot afford to deploy such systems locally. Further, many developers/ users that have little technical know-how and hence rely on querying the available APIs; we thus analyze the most common use-case for using BLOOMz.
>
> **Evaluating other LLMs**: At the time when we ran our experiments, Cohere models were not well optimized for multilingual in-context learning. In preliminary experiments, we saw poor results as the generated responses didn’t reflect any form of instruction following even in higher resourced languages beyond English. Also there was little information shared publicly regarding the multilingual capabilities of Claude, and our focus was on models that were explicitly stated to be multilingual. GPT4 was trained with the same tokenizer as ChatGPT, so we believe that results of our cost analysis will be similar.  We will add more discussion about this in the next version.
>
> **Are fragmentation-rate results generalizable?**: Our results in figure 2 show a trend of languages with non-Latin script having higher fragmentation rates. The majority of the languages that are included in the pretraining data of multilingual LMs are in Latin-script. Our focus here is to therefore highlight the tradeoffs of this data imbalance and how it could affect even higher-resourced languages with non-latin script.
>
> **Summary**: we thank the Reviewer again for their positive and helpful feedback! We will incorporate the above discussions in the final revision.

---

### Meta-Review · Area_Chair_VQPu · 2023-09-22

**Recommendation:** 4

**Metareview:**

*Summary*: It is fairly well-known that the suboptimal tokenization of text across different languages leads to large variance in the number of tokens required to convey the same content in different languages. In this work, the authors systematically analyze the cost and utility of OpenAI's API on 22 typologically diverse languages. The authors find that text in many languages tend to be heavily fragmented, and consequently incur higher costs and derive lower model utility from the OpenAI APIs. This is an analysis-driven paper critiquing the API pricing policy of charging by tokens that puts over-fragmented languages at a clear disadvantage and suggests ways to move forward.

*Evaluations*: R1 and R2 were very favourable in their review of this work (5/4 soundness scores, 4/4 excitement scores). R3 diverges in their review and has listed many concerns including clarifications needed about the main results in both Figs 2 and 3, details about the BLOOM-Z tokenizer, details about the datasets (FLORES corpus, etc.), the more careful use of ``script" whenever invoked, and a few other minor points. The authors have addressed most of these concerns.

---

### Decision · Program_Chairs · 2023-10-07

**Decision:**

Accept-Main

**Comment:**

*Summary*: It is fairly well-known that the suboptimal tokenization of text across different languages leads to large variance in the number of tokens required to convey the same content in different languages. In this work, the authors systematically analyze the cost and utility of OpenAI's API on 22 typologically diverse languages. The authors find that text in many languages tend to be heavily fragmented, and consequently incur higher costs and derive lower model utility from the OpenAI APIs. This is an analysis-driven paper critiquing the API pricing policy of charging by tokens that puts over-fragmented languages at a clear disadvantage and suggests ways to move forward.

*Evaluations*: R1 and R2 were very favourable in their review of this work (5/4 soundness scores, 4/4 excitement scores). R3 diverges in their review and has listed many concerns including clarifications needed about the main results in both Figs 2 and 3, details about the BLOOM-Z tokenizer, details about the datasets (FLORES corpus, etc.), the more careful use of ``script" whenever invoked, and a few other minor points. The authors have addressed most of these concerns.